# Functional Evolution of *Pseudofabraea citricarpa* as an Adaptation to Temperature Change

**DOI:** 10.3390/jof10020109

**Published:** 2024-01-28

**Authors:** Saifei Liu, Li Chen, Xinghua Qiao, Jiequn Ren, Changyong Zhou, Yuheng Yang

**Affiliations:** 1Key Laboratory of Agricultural Biosafety and Green Production of Upper Yangtze River (Ministry of Education), College of Plant Protection, Southwest University, Beibei, Chongqing 400716, China; lsaifei0126@163.com; 2Plant Protection and Fruit Tree Technology Extension Station of Wanzhou District in Chongqing, Chongqing 404199, China; chenli420625@163.com (L.C.); qiaoxh123456@163.com (X.Q.); 3The Chongqing Three Gorges Academy of Agricultural Sciences, Chongqing 404150, China; renjiequn@outlook.com; 4Citrus Research Institute, Southwest University, Beibei, Chongqing 400712, China

**Keywords:** citrus target spot, comparative genomics, high temperature adaptability, virulence evolution

## Abstract

Citrus target spot, caused by *Pseudofabraea citricarpa*, was formerly considered a cold-tolerant fungal disease. However, it has now spread from high-latitude regions to warmer low-latitude regions. Here, we conducted physiological observations on two different strains of the fungus collected from distinct regions, and evaluated their pathogenicity. Interestingly, the CQWZ collected from a low-latitude orchard, exhibited higher temperature tolerance and pathogenicity when compared to the SXCG collected from a high-latitude orchard. To further understand the evolution of temperature tolerance and virulence in these pathogens during the spread process, as well as the mechanisms underlying these differences, we performed genomic comparative analysis. The genome size of CQWZ was determined to be 44,004,669 bp, while the genome size of SXCG was determined to be 45,377,339 bp. Through genomic collinearity analysis, we identified two breakpoints and rearrangements during the evolutionary process of these two strains. Moreover, gene annotation results revealed that the CQWZ possessed 376 annotated genes in the “Xenobiotics biodegradation and metabolism” pathway, which is 79 genes more than the SXCG. The main factor contributing to this difference was the presence of salicylate hydroxylase. We also observed variations in the oxidative stress pathways and core pathogenic genes. The CQWZ exhibited the presence of a heat shock protein (HSP SSB), a catalase (CAT2), and 13 core pathogenic genes, including a LysM effector, in comparison to the SXCG. Furthermore, there were significant disparities in the gene clusters responsible for the production of seven metabolites, such as Fumonisin and Brefeldin. Finally, we identified the regulatory relationship, with the HOG pathway at its core, that potentially contributes to the differences in thermotolerance and virulence. As the global climate continues to warm, crop pathogens are increasingly expanding to new territories. Our findings will enhance understanding of the evolution mechanisms of pathogens under climate change.

## 1. Introduction

Global climate change has been identified as a contributing factor to the increased frequency and severity of fungal disease in plants. The altered climate has disrupted host-pathogen interactions and facilitated pathogen evolution, leading to the emergence of new pathogen strains and escalating the risk of outbreaks [1]. Moreover, fungi are rapidly adapting to the rising temperatures caused by climate change, resulting in an upsurge in pathogenic microorganisms and heightened pathogenicity [2]. Notably, several examples highlight this phenomenon, such as two heat-tolerant strains of *Puccinia striiformis* f. sp. *tritici* [3], the novel race of *P. graminis* f. sp. *tritic* known as Ug99 [4], *Verticillium alfalfae* V31-2 [5], two types of *Fusarium sporotrichioides* and *F. langsethiae* [6]. It is evident that global climate change profoundly influences the evolutionary patterns of pathogens and the interactions between hosts and pathogens, thereby posing significant risks to food safety.

Genetic variation plays a pivotal role in the evolution of adaptive traits [7,8]. Research has demonstrated that the ability of a species to adapt to local environments relies heavily on genetic variation, particularly in polygenic quantitative traits [9]. Fungi exhibit various mechanisms of adaptive genomic differentiation, including (i) amino acid substitutions resulting from mutations in single-nucleotide polymorphisms (SNPs), (ii) alterations in gene copy number or ploidy level, (iii) structural changes in the genome, and modifications in genomic regulatory pathways [10]. Comparative genomic analysis of different strains of the soybean rust revealed that the highly pathogenic strain Race15 has acquired 245 unique genes through evolution [11]. Elevated environmental temperatures induce the activation of the trehalose biosynthetic pathway in corn smut caused by *Ustilago maydis*. This leads to a substantial accumulation of trehalose, which facilitates the pathogen’s adaptation process by enhancing cell permeability protection [12]. Moreover, increasing temperatures pose a substantial threat to vital crops such as potatoes, susceptible to late blight, and rice, vulnerable to blast disease and sheath blight [13]. These studies collectively indicated that the influence of temperature on pathogen local adaptability is multifaceted. Notably, plant pathogens exhibit broader adaptive mechanisms and shorter generation times in comparison to their hosts, allowing them more opportunities to adapt to changing climate conditions. Consequently, new strategies are required to predict and monitor the evolutionary patterns of plant pathogens [14].

Trehalose, heat shock proteins (HSPs), and the antioxidant defense system play significant roles in fungal adaptation to high temperatures [15,16]. Fungal-produced carbohydrate-active enzymes (CAZys) also play a crucial role in the invasion of host plants by efficiently breaking down plant cell walls [17]. Secondary metabolites are essential for the pathogenicity of fungi [18]. The fungal mitogen-activated protein kinase (MAPK) signaling pathway emerges as a critical factor in regulating fungal adaptability to various environmental stresses, including heat tolerance and pathogenicity [19]. Therefore, the mechanisms that fungi have evolved to adapt to high temperatures and a wide range of hosts are complex, multi-layered, and interconnected.

Citrus target spot, caused by *Pseudofabraea citricarpa* was initially discovered as a novel foliar disease in Chenggu County, Shaanxi Province, China. The disease prevailed only in late winter and early spring. It was described as a disease that thrives in low-temperature conditions [20]. Subsequently, in 2019, citrus target spot was also detected in Wanzhou District (Chongqing), Yichang (Hubei Province) and Jishou (Hunan Province) [21]. Our previous studies have demonstrated the critical role of plant cell wall degradation, plant-pathogen protein/polynucleotide interactions, and terpene biosynthesis in the pathogenicity of *P. citricarpa* [22]. Resistance to *P. citricarpa* was evaluated in 27 citrus varieties, with pomelo showing the highest resistance [23]. The geographic expansion of *P. citricarpa* from cooler high latitudes to warmer low latitudes has evolved local adaptations, including adaptation to temperature and to a wider range of hosts.

However, the factors contributing to pathogenic differences in the local adaptation process of *P. citricarpa* remain unclear. To understand the biological changes of *P. citricarpa* during local adaptation, a comparative genomics approach was employed to analyze *P. citricarpa* strains from different latitudes. Our study suggested that involvement of the hyperosmolarity glycerol (HOG) pathway is a significant factor contributing to these differences. This study serves as an example for understanding the impact of climate change on the high temperature tolerance and pathogenicity of fungi. Furthermore, based on these results, specific prevention and control strategies can be developed for different regions aiming to minimize the impact on citrus production within those areas.

## 2. Materials and Methods

### 2.1. Biological Materials

The strains of *Pseudofabraea citricarpa* used in this study were obtained from field surveys. One strain was obtained from Wanzhou District in Chongqing (30°81′ N, 108°61′ E, subtropical monsoon humid zone, annual average temperature of 18.1 °C [24]), while the other was obtained from Chenggu County in Shaanxi Province (33°15′ N, 107°33′ E, transition zone between northern subtropical and warm temperate zones, average annual temperature of 14.2 °C, lower than other citrus-producing areas in China [21]) (Appendix A). The affected citrus leaves were sterilized in a sequential manner with 75% ethanol and 5% sodium hypochlorite. Subsequently, the leaves were rinsed three times with sterile water. The infected leaves were then placed on petri dishes containing potato dextrose agar medium and incubated at 20 °C. Lastly, DNA was extracted from the isolated fungi, and identification was performed.

### 2.2. DNA Extraction

Genomic DNA was extracted using the Omega Fungal DNA Kit D3390-02 (Omega Bio-Tek, Norcross, GA, USA), and the purified genomic DNA was quantified using a TBS-380 fluorometer (Turner BioSystems Inc., Sunnyvale, CA, USA). Only high-quality DNA with an OD_260/280_ ratio of 1.8–2.0 and a quantity greater than 15 μg was utilized for subsequent studies.

### 2.3. Library Construction and Sequencing

The genome was sequenced using a combination of PacBio Sequel Single Molecule Real Time (SMRT) (Pacific BioSciences, Menlo Park, MA, USA) and Illumina (Illumina, San Diego, CA, USA) sequencing platforms (Illumina, San Diego, CA, USA). For Illumina sequencing, at least 5 μg genomic DNA was utilized for sequencing library construction for each strain. The DNA samples were fragmented into 400–500 bp fragments using a Covaris M220 Focused Acoustic Shearer (Covaris, Woburn, MA, USA). The sheared fragments were subsequently utilized to generate Illumina sequencing libraries using the NEXTflex™ Rapid DNA-Seq Kit (Bioo Scientific, Austin, TX, USA). The 5′ ends of the fragments were first end-repaired and phosphorylated, while the 3′ ends were A-tailed and ligated to sequencing adapters. The adapter-ligated products were subsequently enriched through PCR. Subsequently, the prepared libraries underwent paired-end Illumina sequencing (2 × 150 bp) using the Illumina HiSeq X Ten machine (Illumina, San Diego, CA, USA).

The aliquot of 8 μg DNA was spun at 6000 rpm/min for 60 s. The DNA fragments were purified and end-repaired. The resulting sequencing library was purified three times using 0.45 times the volume of Agencourt AMPure XP beads (Beckman Coulter Genomics, Brea, MA, USA). Finally, an ~10 kb insert library was prepared and sequenced.

### 2.4. Genome Assembly Genome Annotation

The genome sequence was assembled using PacBio (Pacific BioSciences, Menlo Park, MA, USA) and Illumina reads (Illumina, San Diego, CA, USA). The initial image data was converted into sequence data via base calling, and saved as FASTQ files. Quality trimming was performed by statistical analysis of the quality information. The contigs were subsequently assembled using CANU v2.1.1. lastly, error correction of the PacBio assembly results was performed using the Illumina reads.

The identification of predicted coding sequences (CDS) was performed by Maker2 v2.31.9. Furthermore, tRNA prediction was performed using the tRNA-scan-SE v2.0 tool, whereas rRNA prediction was conducted with the Barrnap v0.8 software. To annotate the predicted CDSs, sequence alignment tools including BLAST v2.3.0, Diamond v0.8.35, and HMMER v3.1b2 were utilized to align the query proteins with databases such as GO v2.5 and KEGG Latest Version. In summary, the gene annotation was performed by selecting the best-matched subjects with an E-value < 10^−5^.

### 2.5. Comparative Genomic Analysis

The greater the evolutionary distance between species, the worse the genetic collinearity, so the degree of collinearity between two species can be used as a yardstick to measure the evolutionary distance between them. We used Sebelia v3.0.6 to perform collinearity analysis, and then used Circos v0.69-6 to draw the collinear coil diagram.

To analyze the presence of genes related to virulence, PHI genes, secreted proteins, and CAZy in the sequenced. The predicted protein sequences were compared with the Database of Virulence Factors (DFVF) v6, the Pathogen-Host Interaction (PHI) Database v4.4, the SignalP Database v4.1, and the Carbohydrate Active Enzymes (CAZy) Database v6. The information of the database was supplemented in Appendix A.

### 2.6. Statistical Analysis

All experiments were performed using three replicates of biological samples. Colony diameter was measured using the cross method. In the inoculation experiment, a puncture was used to inoculate the leaves from the underside, and the leaves were humidified and cultured at 10 °C. After 30 days, the diameter of the lesions was measured. Data are expressed as mean ± standard deviation and analyzed using SPSS v16.0. Differences between two strains were evaluated using Student’s *t* test. * Significant at *p* < 0.05, ** Significant at *p* < 0.01, *** Significant at *p* < 0.001.

## 3. Results

### 3.1. CQWZ Exhibits Higher Temperature Tolerance and Pathogenicity

Two *P. citricarpa* were found and isolated in the early stages of our investigation. One strain was obtained from Wanzhou District in Chongqing, while the other was obtained from Chenggu County in Shaanxi Province. Chenggu County in Shaanxi Province is the northernmost citrus-producing area in China. The average temperature in the area is 14.2 °C, which is lower than other citrus-producing areas [20,21]. The annual average temperature in Wanzhou District, Chongqing is 18.1 °C. These strains were temporarily named CQWZ and SXCG, respectively. These two strains displayed distinct morphological characteristics on PDA medium while both exhibiting filamentous growth. CQWZ had a white colony color, while SXCG had a dark brown color (Figure 1a). Meanwhile, our study revealed significant differences in the growth rates of two strains. Particularly, CQWZ exhibited significantly higher growth rates at temperatures of 10 °C, 20 °C, and 30 °C compared to SXCG (Figure 1b and Appendix A). Within the first 10 days, the relative growth rate of strain CQWZ at 30 °C was higher than that at 10 °C and 20 °C (Figure 1c). Furthermore, in experiments conducted on major citrus varieties, CQWZ showed greater pathogenicity than SXCG (Figure 1d,e). These findings clearly indicated that the *P. citricarpa* has evolved greater temperature tolerance and pathogenicity as it spreads from high-latitude regions to low-latitude regions.

### 3.2. P. citricarpa Experienced Genome Fragmentation and Fusion during Spread

In this study, the genome of SXCG was sequenced using the same methodology, and two strains genotypes were constructed (Appendix A) [25]. In this study, the genome of SXCG was sequenced using the same methodology, and two strains genotypes were constructed (Appendix A). Comparative analysis between the two strains identified genetic variations. The genome size of SXCG was 45,377,339 bp, containing 13 rRNA genes and 215 tRNA genes, which exceeds that of CQWZ. However, the GC content was higher in CQWZ compared to the SXCG (Table 1 and Appendix A). Furthermore, the number of CDS differed between the two strains (Appendix A). To explore evolutionary relationships, the phylogenetic tree was constructed using genomic data, including several housekeeping genes such as *GAPDH*, *EF-1α*, *β-tubulin*, *Actin*, and *RPL13*. The tree included nine classes of fungi within the Ascomycotina, including highly pathogenic species. Notably, our results indicated that *Sclerotinia sclerotiorum* and *Colletotrichum gloeosporioides* were closely related to *P. citricarpa* (Figure 2a). Collinearity analysis on two strains revealed that the genomes of these strains experienced both fragmentation and fusion events (Figure 2b). In conclusion, these results indicated a close phylogenetic relationship between *P. citricarpa* and *C. gloeosporioides* which also causes citrus diseases. Moreover, fragmentations and fusions of genomic segments have occurred in *P. citricarpa* during its spread.

### 3.3. Differences between Two Strains in Terms of Growth and Pathogenicity Pathways

After predicting the coding DNA sequences of two strains, their biological functions were analyzed using databases, and further functional annotations were performed. Figure 3 showed significantly different numbers of annotated genes in both “carbohydrate metabolism” and “xenobiotics biodegradation and metabolism” pathways. Specifically, CQWZ exhibited 376 annotated genes in the “xenobiotics biodegradation and metabolism” pathway, which is 79 more than that of SXCG. This disparity can be attributed to the salicylic acid hydroxylase. In the “Carbohydrate metabolism” pathway, CQWZ had 641 genes while SXCG had 704 genes. The primary difference in gene composition between the two strains is in the “starch and sucrose metabolism” pathway. Among the three pathways, namely the “digestive system”, “transport and catabolism”, and “immune systems”, strain CQWZ possesses a larger number of annotated genes compared to strain SXCG. Specifically, strain CQWZ has an excess of 3 annotated genes in the “digestive system”, 6 in the “transport and catabolism” pathway, and 5 in the “immune systems” pathway. Additionally, the results from the GO database revealed the absence of genes associated with growth (GO:0040007) and immune system processes (GO:0002376) in SXCG (Figure 3). Notably, the presence of salicylic acid hydroxylase in CQWZ indicated its enhanced capability to break down salicylic acid in citrus, contributing to its stronger pathogenicity compared to SXCG. These observations align with our aforementioned results, which showed that the CQWZ displayed higher levels of pathogenicity and growth. These pathways are potential mechanisms responsible for differences in pathogenicity, growth, and immune systems between the two strains.

### 3.4. Differential High Temperature Tolerance Involves the Oxidative Stress Pathway

To elucidate the reasons for the differences in temperature adaptability and pathogenicity between CQWZ and SXCG. The HSPs, peroxisome, and MAPK signaling pathways were investigated. Our research revealed a distinct HSP (called SSB) in CQWZ compared to SXCG. Furthermore, variations in the amino acid sequences of HSP2 were found between the two strains. While there were no significant differences in peroxidase (POD) between the two strains, disparities were identified in the levels of catalase (CAT) and superoxide dismutase (SOD). Specifically, CQWZ showed the presence of a “putative peroxisomal catalase protein” and differences in the amino acid sequences of SOD2 compared to the SXCG (Figure 4). Our research has also observed differences in the amino acid sequences of proteins Rom1/2, Rho1, and Rlm1 in the cell wall integrity (CWI) pathway, and Sln1, Ssk1, Pbs2, and Hsl7 in the HOG pathway, as well as the associated points Ptp2/3, in the MAPK signaling pathways (Appendix A). Taken together, differences in HSPs and oxidative stress pathways may play crucial roles in the variations observed in thermal adaptation. 

### 3.5. CQWZ Has Evolved to Acquire an Increased Number of Core Pathogenic Genes

Pathogen-host interaction (PHI) genes, carbohydrate-active enzymes (CAZyme), and secreted proteins (SP) are considered core pathogenic factors. In CQWZ, a total of 1280 secreted proteins were predicted, which was 94 more than SXCG. The genome annotation of CQWZ showed a larger number of CAZymes, including glycoside hydrolases, glycosyl transferases, polysaccharide lyases, carbohydrate esterases, and others (Figure 5a). Compared to SXCG, CQWZ possessed a unique effector similar to SLP1 of *Magnaporthe oryzae* (Table 2 and Appendix A). The Venn diagram analysis of PHI genes, CAZymes, and SPs determined that CQWZ exhibited 11 additional pathogenic factors, including well-known fungal pathogenic factors such as GMC oxidoreductase, Cutin hydrolase 1, xyloglucanase, Lipase 3, alcohol oxidase, and Pectate lyase E (Figure 5b and Appendix A). The presence of these factors enables CQWZ to penetrate plant tissues and destroy plant cell walls more effectively.

The pathogenicity of fungi was greatly aided by the synthesis of secondary metabolites. Therefore, it is crucial to estimate the secondary metabolite gene clusters of the two strains. Gene clusters were accurately predicted using antiSMASH v5.1.2, and differences between the secondary metabolite gene clusters were determined. The existence of gene clusters encoding various fungi toxins, including fumonisin and nivalenol, was one of these variances. Major differences were also identified in gene clusters involved in the synthesis of additional secondary metabolites (Figure 5c). Variations in secondary metabolite synthesis gene clusters contribute to the variation in pathogenicity between the two strains. In addition, drug-resistance genes were predicted for the two strains, and more resistance genes were found in SXCG than in CQWZ, particularly stronger resistance against Macrolide (Appendix A). The heightened pathogenicity displayed by CQWZ can be attributed to the expansion of core pathogenic factors and changes in the gene clusters accountable for fungal toxin synthesis.

### 3.6. HOG Pathway Responsible for Differences in High Temperature Tolerance and Pathogenicity

To understand the key factors contributing to the enhanced heat resistance and pathogenicity of CQWZ, a comprehensive analysis was conducted. This analysis revealed four factors associated with high-temperature tolerance and pathogenicity, namely the MAPK pathway, pyruvate metabolism, terpenoid backbone biosynthesis, and sphingolipid metabolism (Figure 6). Variations in the sequences of essential genes involved in the sphingolipid metabolism pathways, including KDSR, SPHK, ASAH2, and SMPD2, as well as those genes in the terpenoid skeleton biosynthesis pathway like DXS, were observed. There were also differences in genes associated with pyruvate metabolism, such as pckA, LDH, and ldD, and a regulatory relationship exists between these four pathways. The sphingolipid pathway regulates Pkc1 through the formation of diacylglycerol [26], acetate induces activation of Hog1 [27], the first step of the nomevalonate pathway involves the synthesis of 1-deoxy-d-xylulose 5-phosphate (DXP) through the condensation of pyruvate and glyceraldehyde 3-phosphate via DXP synthase catalysis [28].

## 4. Discussion

Climate change has significant impacts on the growth and development of phytopathogens, as well as on the immune systems of host organisms and the physiological processes involved in their interaction. These changes have the potential to impact the geographic distribution of pathogens, resulting in crop losses [29]. Previous studies have primarily relied on climate change models to predict the spread and evolution of fungal pathogens. Here, we present a case study of *P. citricarpa*, which demonstrates the migration of the pathogen from colder high latitudes to warmer lower latitudes, enabling it to invade a wider range of citrus crops. The use of *P. citricarpa* as an empirical model allows for the study of evolutionary molecular plant-microbe interactions in the context of climate change.

### 4.1. Temperature Adaptation of Fungi

Fungi exhibit thermal adaptations that are closely associated with specific traits, such as growth rate and morphology. The genetic mechanisms underlying these adaptations can be complex, as demonstrated by *Zymoseptoria tritici*, which can adapt rapidly to changing environmental conditions and thereby expand its range into new climatic regions [30]. To survive in dynamic environments, fungi have developed various adaptive behaviors, including heat response systems, conserved signaling pathways, transcriptional regulatory systems, essential physiological and biochemical processes, and notable phenotypic changes [31].

To adapt to high-temperature environments, fungi employ various strategies, including the activation of heat shock transcription factors (HSFs), antioxidant responses, heat shock responses, and the accumulation of trehalose [32]. Trehalose-6-phosphate synthase 1 (TPS1) is a key enzyme involved in trehalose synthesis in fungi [33]. Our genomic study showed significant differences in the TPS1 sequence between the two strains. Fungi also utilize sphingolipids (SLs) as temperature sensors to regulate the activity of lipid molecules in the membrane and enzymes in the SL pathway in response to thermal stresses. These progressions can activate intracellular signals and work with the variation of parasites to temperature changes [34]. Additionally, fungal adaptation to high-temperature stress is characterized by an increased accumulation of 9(11)-dehydroergosterol and ergosterol peroxide [35]. In our study, significant differences in gene sequences encoding key enzymes in the sphingolipid metabolism pathway were observed, which is one of the potential factors leading to high temperature tolerance in strain CQWZ. 

When exposed to higher temperatures, cells experience increased oxygen respiration rates, leading to the accumulation of reactive oxygen species (ROS) and subsequent cellular damage [36]. Fungi respond to oxidative stress through the regulation of various pathways, including antioxidant enzymes and pyruvic acid. Under heat stress conditions, fungi utilize pyruvic acid as a primary strategy to resist heat-induced ROS [37]. Our study demonstrated significant differences in the sequences of key genes of the pyruvate metabolic pathway and ROS scavenging system between the two strains. Fungi possess complex mechanisms to adapt to elevated temperatures, including HSPs, trehalose pathway, MAPK pathway, and pyruvic acid pathway, all of which are important potential factors leading to high temperature tolerance among strains.

### 4.2. Fungal Manipulation of Plant Defenses

Rising temperatures pose a threat to agricultural productivity by promoting the rapid reproduction and spread of phytopathogens. To protect crops, it is essential to adopt innovative approaches that integrate fungal genomics into plant pathology research [38]. This integration will enhance our understanding of the intricate interactions between fungal pathogens and host crops, providing insights and strategies for disease prevention and management. Additionally, this study could further our understanding of the evolutionary dynamics of fungi under global warming conditions.

Temperature plays a crucial role in modulating the virulence mechanisms of pathogens, including synthesis of toxins and virulence proteins, pathogen replication, and overall survival [39]. These findings indicated that during local adaptation of strain CQWZ, adaptation to high temperatures and a wide range of hosts are related. During evolution, the strain CQWZ has effectively adapted to diverse ecological environments and displayed strong pathogenicity. One-way pathogens achieve this adaptive capability is through effectors that disrupt plant defense signaling pathways or mimic specific molecules to evade host immune responses. Fungi often target and destroy plant defense hormones, such as ethylene, salicylic acid (SA), and jasmonic acid, during invasion [40]. For instance, *U. maydis* produces a salicylate hydroxylase Shy1, which breaks down SA in the host plant, thereby promoting fungal invasion [41]. Our study found that CQWZ possesses a higher number of salicylate hydroxylase enzymes compared to SXCG, indicating its robust capacity to degrade host endogenous SA. 

*Pseudofabraea citricarpa* is capable of surviving inside its host, it does not belong to the category of necrotrophic fungi, and effectors play an important role in its virulence. Our findings indicated that the strain CQWZ secretes an additional effector containing two lysin-motif (LysM) domains compared to SXCG (Table 2). Fungal LysM effectors disrupt chitin-activated immune responses in plants, and examples include Ecp6, Mg1LysM, Mg3LysM, and Mgx1LysM [42,43]. The LysM effector in our study showed similarity to the secreted effector Slp1 of *Magnaporthe oryzae* recorded in the PHI database. Slp1 plays a crucial role in the pathogenicity of *M. oryzae* by competing with OsCEBiP for chitin binding and inhibiting chitin-triggered plant immune responses [44]. Another chitin receptor called lysin-motif receptor like kinase family (LYKs) exists in citrus, but CEBiP has not been identified [45]. Therefore, the LysM effector may compete with LYKs for binding to chitin, and disrupt chitin-triggered immune responses. Additionally, both strains contain homologs of PsXEG1, which play a critical role in the interaction between soybeans and *Phytophthora sojae* [46]. Similar mechanisms may exist in the interaction between citrus and *P. citricarpa*. Our research also revealed differences in CAZys between the two strains, with the strain CQWZ possessing a higher number of core pathogenic genes compared to the strain SXCG.

## Figures and Tables

**Figure 1 jof-10-00109-f001:**
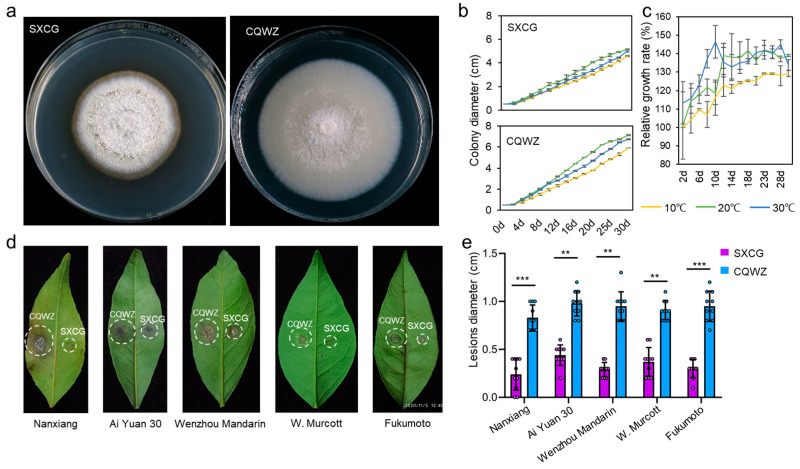
Two strains of *Pseudofabraea citricarpa* differ in morphology and pathogenicity. (**a**) Compared the colony morphology of two different strains of fungi from different sources. The colony pictures were taken when the CQWZ strain had grown to two-thirds of the petri dish. (**b**) Comparison of the growth rates of the two strains at 10 °C, 20 °C and 30 °C. (**c**) Relative growth rates of strain CQWZ at three temperatures relative to that of strain SXCG. (**d**) Differences in pathogenicity between two strains of *P. citricarpa* on several citrus varieties. (**e**) Lesion diameter statistics, ** *p* < 0.01, and *** *p* < 0.001.

**Figure 2 jof-10-00109-f002:**
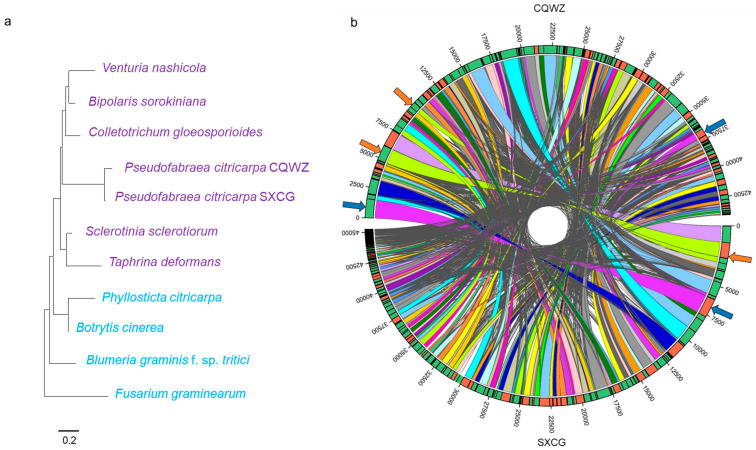
Evolutionary relationships of *Pseudofabraea citricarpa.* (**a**) Maximum likelihood trees of *Pseudofabraea citricarpa*. (**b**) Genomic colinearity analysis between SXCG and CQWZ. The colored regions represent the areas which are aligned in the sequence alignment. Each line represents an alignment record. If the alignment is in the forward direction, the outer ring of the alignment region is the same color. If the alignment is in the reverse direction, the outer ring color is different.

**Figure 3 jof-10-00109-f003:**
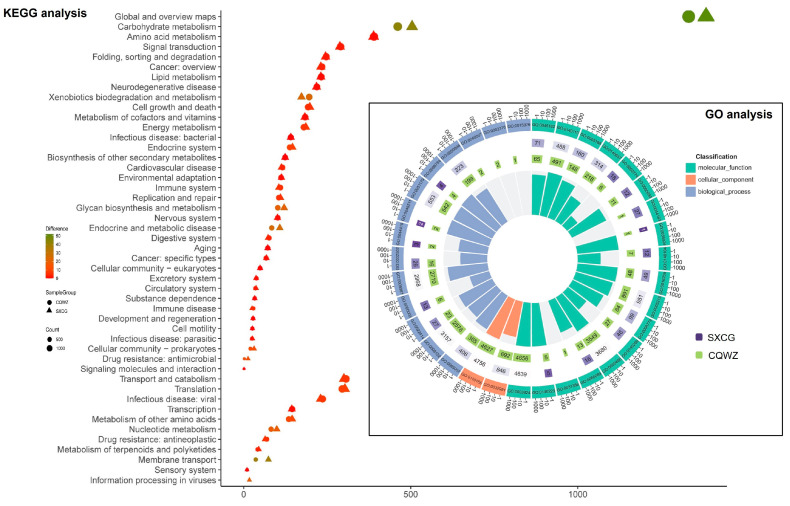
Differences in gene annotation between two strains. Differences in gene annotation between two strains in the KEGG database. Circles represent CQWZ strains and triangles represent SXCG strains. Change from red to green indicates the difference in the number of genes between the two strains. Differences in gene annotation between the two strains in the GO database. The numbers in the inner circle represent the number of genes in the two strains. The bars indicate the difference in the number of genes in the pathway between the two strains. The higher the bar, the less obvious the overall difference in the number of genes between the two strains. The absence of a bar means that a certain strain has no genes in this pathway.

**Figure 4 jof-10-00109-f004:**
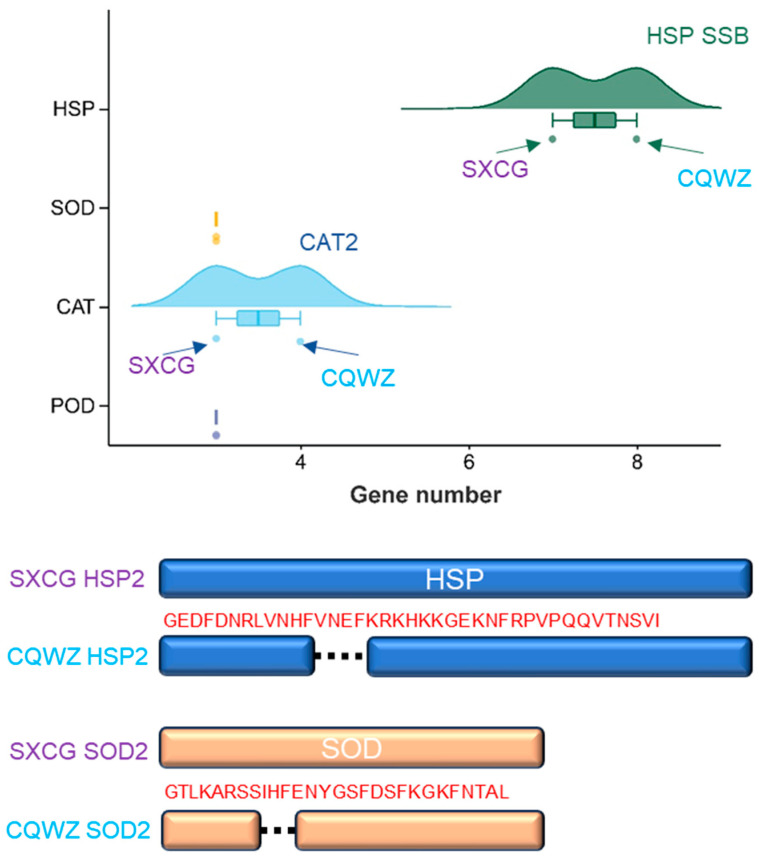
Mechanisms of high temperature adaptation differences between two strains. Comparison of the quantity and amino acid sequence of HSP, SOD, CAT, and POD in two different strains.

**Figure 5 jof-10-00109-f005:**
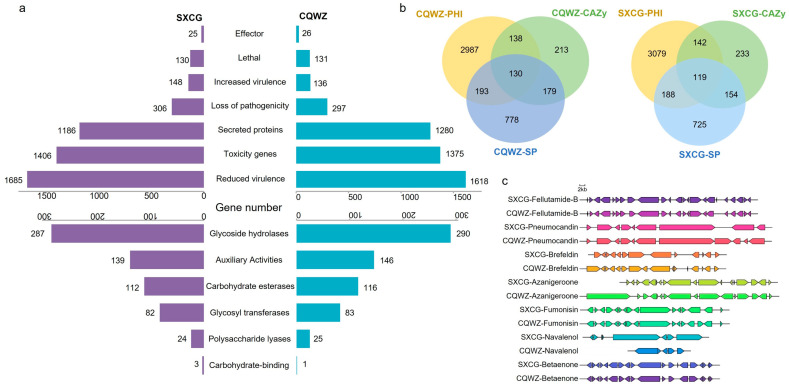
Mechanisms of difference in the pathogenic systems of the two strains. (**a**) Differences in the number of pathogenesis-related genes in SXCG and CQWZ. Purple indicates SXCG strains, and blue indicates CQWZ strains (**b**) Screening for core virulence factors of SXCG and CQWZ. (**c**) Differences in gene clusters for secondary metabolite synthesis.

**Figure 6 jof-10-00109-f006:**
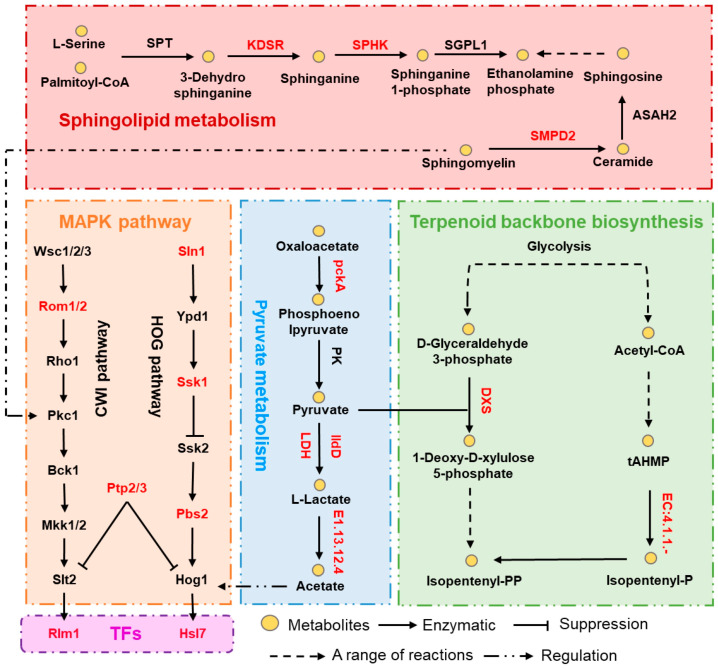
Combined mechanisms of high-temperature adaptation and pathogenesis-related pathways. Red text indicates differences in amino acid sequences between the two strains.

**Table 1 jof-10-00109-t001:** Basic information on the genomes of the two strains.

Features	Chromosome	
	CQWZ	SXCG
Genome Size (bp)	44,005,176	45,377,339
GC Content (%)	44.71	44.54
CDS No.	14,193	14,110
rRNA No.	4	13
tRNA No.	163	215

**Table 2 jof-10-00109-t002:** The predicted effectors of two strains of fungi based on genome sequences.

PHI ID	Protein ID	Homologous Name	NCBI Tax ID	Pathogen Species	Number	Sample
PHI:7376	J4URT3	Blys2	176275	*Beauveria bassiana*	1	SXCG/CQWZ
PHI:6833	G2XA95	Vd6LysM	27337	*Verticillium dahliae*	2	SXCG/CQWZ
PHI:9334	D2TI55	map	67825	*Citrobacter rodentium*	2	SXCG/CQWZ
PHI:2216	A0N0D1	PemG1	318829	*Magnaporthe oryzae*	1	SXCG/CQWZ
PHI:5495	B3VBK9	Ecp6	5507; 5499	*Passalora fulva*; *Fusarium oxysporum*	3	SXCG/CQWZ
PHI:7664	A0A0A2ILW0	LysM1	27334	*Penicillium expansum*	2	SXCG/CQWZ
PHI:7667	A0A0A2JB06	LysM4	27334	*P. expansum*	1	SXCG/CQWZ
PHI:2404	G4N906	SLP 1	318829	*M. oryzae*	1	CQWZ
PHI:3216	G4MVX4	MoCDIP4	318829	*M. oryzae*	3	SXCG/CQWZ
PHI:3213	G4N8Y3	MoCDIP1	318829	*M. oryzae*	1	SXCG/CQWZ
PHI:5335	A0A0H3HVK0	clpV-5	28450	*Burkholderia pseudomallei*	1	SXCG/CQWZ
PHI:325	Q6ZX14	ACE1	318829	*M. oryzae*	5	SXCG/CQWZ
PHI:6868	G4ZHR2	PsXEG1	67593	*Phytophthora sojae*	2	SXCG/CQWZ
PHI:981	Q8RP09	hopI1	317	*Pseudomonas syringae*	1	SXCG/CQWZ

## Data Availability

All data supporting the findings of this study are available within the paper and within its Appendix A published online. The genomic information of two strains (SXCG-NMDC60134480, CQWZ-NMDC60045940) has been successfully uploaded to the National Microbiology Data Center (NMDC, http://resolve.pid21.cn/13913.11.micro.data.genome.NMDC60134480, accessed on 6 September 2023, http://resolve.pid21.cn/CSTR:13913.11.micro.data.genome.NMDC60045940, accessed on 1 December 2022). Furthermore, the genomic information of two strains (SXCG-SUB13926630 Processing, CQWZ-JAPWGK000000000) has been successfully uploaded to the National Center for Biotechnology Information (NCBI, https://www.ncbi.nlm.nih.gov/ accessed on 20 November 2023).

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
