# Peer review of "Functional Evolution of Pseudofabraea citricarpa as an Adaptation to Temperature Change"

_jof, 2024, doi:10.3390/jof10020109_

Round 1

Reviewer 1 Report

Comments and Suggestions for Authors

Dear authors,

Here are my comments about the manuscript:” Functional evolution of Pseudofabraea citricarpa as an adaptation to climate change”

The manuscript is clear and well written and doesn’t need English editing. Just minor changes. I will label the ones I find/see. In general, it is good and if the data hold true, I think it presents valuable information.

However, my major concern is regarding the selected strains, the data and how the data are presented here. In my opinion the strains don’t show a difference in growth in relation to temperature, rather a general growth difference between the 2 strains. Thus, the data have to be reanalyzed, with standard deviation and statistics to show a difference in growth depending on temperature for the 2 strains. If this cannot be shown the whole manuscript collapses regarding the drawn conclusions.  

Regarding the data:

Selection of strains: The strains came from 2 regions roughly (3 degrees latitude apart) 350km apart regarding north south distant? How was the elevation difference between those regions (for example if the southern region is higher in elevation compared to the more northern region, the climate (temperature) of the 2 regions could actually be identical!)

Figure 2 b: It appears that strain SXCG is growing slower at every tested temperature at every day compared to CQWZ. Please provide a table which shows for each data point and for each temperature the percentage of growth between the 2 strains. Also include standard deviation and statistics here. They are essential for the rest of the story as it is the foundation!

What I’m getting at is, if strain CQWZ has a relative growth rate of 150 percent compared to SXCG at every tested timepoint for 10 degrees and 20 degrees or 30 degrees, it would only show that this strain grows better in general. If you can show that at 10 degrees it is always 150 percent (for example) at 20 degrees the difference is 180 percent growth rate and at 30 degrees 250 percent. This would show that CQWZ indeed can cope better with higher temperatures. At the moment with the provided data and without standard deviation and statistics it appears that CQWZ in general grows better and has not adapted to higher temperatures as suggested. I have estimated the relative growth rate of CQWZ from the graph at day 30 to be around 138 percent at 10 degrees, 140 percent at 20 degrees and 129 percent at 30 degrees compared to SXCG (unfortunately).

This brings me to the infection assay. If CQWZ grows faster, of course the infection will spread faster just because of this growth difference. Which would not indicate a difference in virulence but rather growth rate in general again.

Also, regarding effectors as seen in table 2, if this is a necrotrophic fungus, effectors are not very helpful during infection, if it is a hemibiotroph it should be said somewhere in the text.

Also, in the materials and methods section there are no information about growth and infection assays to be found or which stats methods were used. For example is the infection assay at 30 degrees, for 1 day or 30 days,…. These are important information.

The GC content between the strains is 44.71 and 44.54 percent. This shows no differences between the 2 strains. Maybe you could highlight sections of the genome with significantly different GC content, which would indicate horizontal gene transfer between different organism/species for those sections.

Figure 2: is a and b switched between graph and description? and in a which colors are used to indicate directions should be named in the description,…  (further, green and red are not good options as not every person is able to perceive those)

Figure 4: are the gDNA sequences missing in the genome sequence for CQWZ (hsp2 and sod2)? Or are they there, but software called for different intron exon splicing variants between the 2 strains? If this is the case, were the different isoforms confirmed with differences in cDNA sequences for SXCG and CQWZ?

Fig 5: in a the gene number labelling for the lower part of this graph is upside down, please fix it. In 2 b it is called Navalenol while in the text line 268 nivalenol.

For the pathway analysis and the putative impact on growth, temperature and virulence. All the data are only related to bioinformatic analysis, correct? Because I don’t see any assays to confirm for example trehalose concentrations are different between the strains or pyruvate concentrations, or deletion strains for any of the identified putative functions to confirm the conclusions.  If this is the case all those sections should indicate that it is putative and only because of bioinformatic analysis. Is there correlation or causation?!? Between the seen differences between the 2 strains?  Thus, please tune it down.

Line 111: and he purified; change to the   

Line 206 is repeated in line 207

Line 275: maybe better expansion rather than escalation

Author Response

Dear Editor,

We have made changes based on your requests and the reviewers' requirements. Now resubmitting our manuscript (ID: jof-2770688-R1) to Journal of Fungi.

Thank you very much for reviewing our manuscript and giving us your valuable comments! We have carefully revised our manuscript according to journal requirements and your comments. We have thoroughly reviewed the manuscript and made necessary revisions according to your feedback. Thanks again for the careful review by the editor.

We have reviewed the final version of the manuscript and approve it for publication. To the best of our knowledge and belief, this manuscript has not been published in whole or in part nor is it being considered for publication elsewhere. The authors declare that they have no competing interests.

Many thanks for your time and consideration of this manuscript. We look forward to hearing about your comments/decision.

Kind regards,

Prof. Yuheng Yang

College of Plant Protection

Southwest University

Chongqing, China, 400716

Reviewer 1:

Here are my comments about the manuscript:” Functional evolution of Pseudofabraea citricarpa as an adaptation to climate change”

The manuscript is clear and well written and doesn’t need English editing. Just minor changes. I will label the ones I find/see. In general, it is good and if the data hold true, I think it presents valuable information.

However, my major concern is regarding the selected strains, the data and how the data are presented here. In my opinion the strains don’t show a difference in growth in relation to temperature, rather a general growth difference between the 2 strains. Thus, the data have to be reanalyzed, with standard deviation and statistics to show a difference in growth depending on temperature for the 2 strains. If this cannot be shown the whole manuscript collapses regarding the draw conclusions.  

Response: We thank the reviewers for their positive comments and critical and constructive suggestions. We believe that the reviewers raised important points and these comments allowed us to further clarify and strengthen the manuscript. In the revised manuscript, we re-performed the statistical analysis of the data. The results clearly showed that under conditions of 10, 20, and 30°C, the colony diameter of strain CQWZ was significantly higher than that of SXCG after 30 days of culture.

Regarding the data:

Selection of strains: The strains came from 2 regions roughly (3 degrees latitude apart) 350km apart regarding north south distant? How was the elevation difference between those regions (for example if the southern region is higher in elevation compared to the more northern region, the climate (temperature) of the 2 regions could actually be identical!)

Response: The sampling location of strain SXCG is the northernmost and highest altitude main citrus producing area in China. The annual average temperature is 14.1 °C. The sampling site of strain CQWZ has a humid subtropical monsoon climate; the annual average temperature is 18.2 °C. The data comes from the region's official government website.

Figure 2b: It appears that strain SXCG is growing slower at every tested temperature at every day compared to CQWZ. Please provide a table which shows for each data point and for each temperature the percentage of growth between the 2 strains. Also include standard deviation and statistics here. They are essential for the rest of the story as it is the foundation!

What I’m getting at is, if strain CQWZ has a relative growth rate of 150 percent compared to SXCG at every tested timepoint for 10 degrees and 20 degrees or 30 degrees, it would only show that this strain grows better in general. If you can show that at 10 degrees it is always 150 percent (for example) at 20 degrees the difference is 180 percent growth rate and at 30 degrees 250 percent. This would show that CQWZ indeed can cope better with higher temperatures. At the moment with the provided data and without standard deviation and statistics it appears that CQWZ in general grows better and has not adapted to higher temperatures as suggested. I have estimated the relative growth rate of CQWZ from the graph at day 30 to be around 138 percent at 10 degrees, 140 percent at 20 degrees and 129 percent at 30 degrees compared to SXCG (unfortunately).

Response: We appreciate it very much for this good suggestion, and we have done it according to your ideas. We have supplemented the data in the attached table, calculating the relative growth rate of strain CQWZ at each temperature at each time point relative to strain SXCG. We understand the reviewer's point. On the 30th day, the relative growth rate at 30 °C was lower than 20 °C. The optimal growth temperature of Pseudofabraea citricarpa is 20 °C. And the period of our measurement is longer. The PDA medium may not be sufficient for the growth of the fungus at 30 days, but at other time points, the growth rate of strain CQWZ at 30 °C is much higher than other temperature conditions.

This brings me to the infection assay. If CQWZ grows faster, of course the infection will spread faster just because of this growth difference. Which would not indicate a difference in virulence but rather growth rate in general again. Also, regarding effectors as seen in table 2, if this is a necrotrophic fungus, effectors are not very helpful during infection, if it is a hemibiotroph it should be said somewhere in the text. Also, in the materials and methods section there are no information about growth and infection assays to be found or which stats methods were used. For example, is the infection assay at 30 degrees, for 1 day or 30 days. These are important information.

Response: We thank the reviewer for the comments and feedback. Our previous study showed that the strain originating from Chongqing has a wider range of hosts and can infect a variety of citrus and pomelo species, suggesting that it has evolved higher virulence. Pseudofabraea citricarpa is capable of surviving inside its host, it does not belong to the category of necrotrophic fungi, as we have noted in line 308 of the manuscript. For infection experiments, we used a system in which acupuncture was applied to the leaf back and moisturized, and cultured at 10 °C for 28 days. Moreover, we have also supplemented the statistical methods in the Materials and Methods section of the new manuscript.

The GC content between the strains is 44.71 and 44.54 percent. This shows no differences between the 2 strains. Maybe you could highlight sections of the genome with significantly different GC content, which would indicate horizontal gene transfer between different organism/species for those sections.

Response: We very much appreciate the reviewer’ comment. We also considered the ideas proposed by the reviewer, but found it difficult to present them graphically. Therefore, we supplemented Figure S2 and Figure S3 to show the difference in GC content of the two strains.

Figure 2: is a and b switched between graph and description? and in which colors are used to indicate directions should be named in the description, (further, green and red are not good options as not every person is able to perceive those)

Response: We are very grateful to Reviewer for reviewing the paper so carefully. We have made changes to Figure 2 and described them in the corresponding section of the article.

Figure 4: are the gDNA sequences missing in the genome sequence for CQWZ (hsp2 and sod2)? Or are they there, but software called for different intron exon splicing variants between the 2 strains? If this is the case, were the different isoforms confirmed with differences in cDNA sequences for SXCG and CQWZ?

Response: Compared with strain SXCG, part of the HSP2 and SOD2 sequences of strain CQWZ was deleted, and the cDNA sequences of the two strains also showed differences.

Fig 5: in the gene number labelling for the lower part of this graph is upside down, please fix it. In 2 b it is called Navalenol while in the text line 268 nivalenol.

Response: We gratefully acknowledge the reviewer's suggestion and we have made changes to Figure 5. We are very sorry for our incorrect writing and it is rectified in Line 268.

For the pathway analysis and the putative impact on growth, temperature and virulence. All the data are only related to bioinformatic analysis, correct? Because I don’t see any assays to confirm for example trehalose concentrations are different between the strains or pyruvate concentrations, or deletion strains for any of the identified putative functions to confirm the conclusions. If this is the case all those sections should indicate that it is putative and only because of bioinformatic analysis. Is there correlation or causation?!? Between the seen differences between the 2 strains? Thus, please tune it down.

Response: We gratefully acknowledge the reviewers' suggestions. As the reviewer noted, we did not measure relevant metrics. We conducted annotation analysis based on the genome information of the two strains and found differences in several pathways related to temperature tolerance and virulence, including differences in the number of annotated genes, amino acid sequences, etc., so we speculated that these pathways are fungal Mechanisms underlying adaptation to high temperatures and virulence potential. We toned it down in the article. In our future work, we will further confirm whether there are differences in temperature adaptation mechanisms and pathogenicity mechanisms between strains by experimental methods. Now, gene knockout system for Pseudofabraea citricarpa is still in the process of being established.

Line 111: and he purified; change to the

Response: We are very sorry for our incorrect writing and it is rectified in Line 111.

Line 206 is repeated in line 207

Response: We are very sorry for our negligence, and it has been revised.

Line 275: maybe better expansion rather than escalation

Response: “escalation” was replaced by “expansion” in Line 275.

Reviewer 2 Report

Comments and Suggestions for Authors

The paper presents data on Citrus target spot, affecting orchards. The paper contains an exhaustive molecular characterization comparing two strains of the pathogen, one from high latitude regions and other from low latitude regions. However, the paper claims adaptation to climate change, the authors do not show any record of temperature, CO2 or any other metric related to changes in the climate of the regions. The title should be rewritten to avoid confusing the reader.

The data deposited in public repositories contains no usable data. The links given in the manuscript, lead to profiles with no downloadable data. For example, searching in ncbi there are the profiles SAMN32156912 (for CQWZ) and SAMN37942572 (for SXCG) but none of them contain downloadable data. The same occurs for NMDC. Without this information, reproducibility/contrast of the data is impossible. Please, refresh/clarify the information on data availability.

Author Response

Dear Editor,

We have made changes based on your requests and the reviewers' requirements. Now resubmitting our manuscript (ID: jof-2770688-R1) to Journal of Fungi.

Thank you very much for reviewing our manuscript and giving us your valuable comments! We have carefully revised our manuscript according to journal requirements and your comments. We have thoroughly reviewed the manuscript and made necessary revisions according to your feedback. Thanks again for the careful review by the editor.

We have reviewed the final version of the manuscript and approve it for publication. To the best of our knowledge and belief, this manuscript has not been published in whole or in part nor is it being considered for publication elsewhere. The authors declare that they have no competing interests.

Many thanks for your time and consideration of this manuscript. We look forward to hearing about your comments/decision.

Kind regards,

Prof. Yuheng Yang

College of Plant Protection

Southwest University

Chongqing, China, 400716

Reviewer 2:

The paper presents data on Citrus target spot, affecting orchards. The paper contains an exhaustive molecular characterization comparing two strains of the pathogen, one from high latitude regions and other from low latitude regions. However, the paper claims adaptation to climate change, the authors do not show any record of temperature, CO2 or any other metric related to changes in the climate of the regions. The title should be rewritten to avoid confusing the reader.

Response: We are very grateful to Reviewer for reviewing the paper so carefully. We took the reviewer's suggestion seriously and revised our title.

The data deposited in public repositories contains no usable data. The links given in the manuscript, lead to profiles with no downloadable data. For example, searching in ncbi there are the profiles SAMN32156912 (for CQWZ) and SAMN37942572 (for SXCG) but none of them contain downloadable data. The same occurs for NMDC. Without this information, reproducibility/contrast of the data is impossible. Please, refresh/clarify the information on data availability.

Response: We very much appreciate the reviewers’ suggestions. We have uploaded the genome information of these two strains to NMDC (SXCG-NMDC60134480, CQWZ-NMDC60045940), but we did not choose to make it fully public, we chose to make it public under the agreement. Regarding this issue, we have contacted the staff of NMDC, and you will be able to see the genome information once it is resolved. The genome information of strain CQWZ has been uploaded to NCBI database (JAPWGK000000000), but these data have not yet been released due to the long review process at NCBI. We are confident that whether it is NMDC or NCBI, our data will soon be released.

Reviewer 3 Report

Comments and Suggestions for Authors

The authors identified two different strains of the fungi collected from distinct regions, and the CQWZ exhibited higher thermotolerance and pathogenicity when compared to the SXCG. These results showed the evolution mechanisms of pathogens under climate change, as well as provide possible prevention and control strategies for citrus production. In order to further improve the manuscript, I have the following suggestions:

1. Figure 4, Does the amino acid sequence belongs to SOD2 or CAT2? Figure 4 Legend, comparison of the amino acid sequence include only HSP and SOD2.

2. Line 223-225: This sentence from “Specifically, the CQWZ contains 641 genes, while the SXCG contains 704 genes.” is repeated

3. In the previous, e.g. line 161/245/325 is written for temperature tolerance, and in the back, e.g. line 258/298 is written for high-temperature tolerance, Please describe it accurately whether is temperature tolerance or high-temperature tolerance.

4. In line 384-385, the CQWZ strain secretes an additional effector SPL1 is The additional effector ?Table 2 does not show SPL1 contain two lysin-motif (LysM) domains. I suggest adding a schematic diagram of the domain of the SLP1.

Minor suggestion

1. Line 187-190: The main character should be CQWZ.

2. In line 232-234, I suggest add pathways “digestive system”, “transport and catabolism” and “immune systems”in result. The description of KEGG analysis does not reflect the growth and immune systems, only pathogenicity.

Author Response

The authors identified two different strains of the fungi collected from distinct regions, and the CQWZ exhibited higher thermotolerance and pathogenicity when compared to the SXCG. These results showed the evolution mechanisms of pathogens under climate change, as well as provide possible prevention and control strategies for citrus production. In order to further improve the manuscript, I have the following suggestions:

Response: We would like to express our sincere appreciation to the reviewer for your insightful comments and suggestions.  We have carefully reviewed each comment and suggestion made by the reviewer and have made the necessary revisions accordingly.

1.Figure 4, Does the amino acid sequence belong to SOD2 or CAT2? Figure 4 Legend, comparison of the amino acid sequence includes only HSP and SOD2.

Response: We are very grateful to the reviewer for your careful review. We are very sorry for our negligence. In Figure 4, we compared the quantity and sequence differences of HSP, SOD, CAT and POD between the two strains. Among them, HSP and CAT show differences in quantity, and HSP (HSP2) and SOD (SOD2) show differences in sequence.

  1. Line 223-225: This sentence from “Specifically, the CQWZ contains 641 genes, while the SXCG contains 704 genes.” is repeated.

Response: “Specifically, the CQWZ contains 641 genes, while the SXCG contains 704 genes.” was deleted.

  1. In the previous, e.g. line 161/245/325 is written for temperature tolerance, and in the back, e.g. line 258/298 is written for high-temperature tolerance, please describe it accurately whether is temperature tolerance or high-temperature tolerance.

Response: We are very grateful to Reviewer for reviewing the paper so carefully. We have corrected the corresponding parts of the article and made it consistent.

  1. In line 384-385, ‘the CQWZ strain secretes an additional effector’ SPL1 is the additional effector? Table 2 does not show SPL1 contain two lysin-motif (LysM) domains. I suggest adding a schematic diagram of the domain of the SLP1.

Response: We are very grateful for the reviewer's comment, it is a very good suggestion. In our results, CQWZ secretes an additional effector similar to SLP1 secreted by Magnaporthe oryzae in the PHI database. In the newly submitted manuscript, we analyzed the PcSLP1 domain, which is identical to the MoSLP1 domains and contains a signal peptide and two LysM domains (Figure S6).

Minor suggestion

  1. Line 187-190: The main character should be CQWZ.

Response: Changes have been made to the corresponding parts of the article.

  1. In line 232-234, I suggest add pathways “digestive system”, “transport and catabolism” and “immune systems”in result. The description of KEGG analysis does not reflect the growth and immune systems, only pathogenicity.

Response: We greatly appreciate the suggestion provided by the reviewer. “Among the three pathways, namely the "digestive system," "transport and catabolism," and "immune systems," strain CQWZ possesses a larger number of annotated genes compared to strain SXCG. Specifically, strain CQWZ has an excess of 3 annotated genes in the "digestive system," 6 in the "transport and catabolism" pathway, and 5 in the "immune systems" pathway” was added.  

Round 2

Reviewer 1 Report

Comments and Suggestions for Authors

Dear authors,

The strain CQWZ grows at 10 degrees in the range of 100-129% better than SXCG (lowest and highest value from all sample points). At 20 degrees 89-142% and at 30 degrees 113-146%. Over all sample data points this makes an average of 119%, 129% and 134% relative growth for 10 degree, 20 and 30, respectively (according to Table S1). Thus, there is only a very small difference (less than 5%) between 20 and 30 degrees of growth improvement of CQWZ over SXCG at 30 degrees. Additionally at 7 out of the 14 data points the relative growth improvement of CQWZ is highest at 20 degrees not at 30 degrees. Thus, there is no relative growth difference regarding temperature (at least at 20 vs 30) between the 2 strains.

Considering the small differences of the average and the variability of the data including measurement variability of 0.43 up to 18.3 % at the different time points. I don’t think there is a significant better growth of CQWZ at 30 degrees compared to 20 degrees and relative to SXCG.

I’m afraid without additional strains showing that the SXCG strain and their growth pattern at different temperatures actually represents the majority of the population and CQWZ is a significant outlier regarding temperature tolerance the presented data don’t warrant the conclusion of the manuscript.       

Comments on the Quality of English Language

is fine

Author Response

The strain CQWZ grows at 10 degrees in the range of 100-129% better than SXCG (lowest and highest value from all sample points). At 20 degrees 89-142% and at 30 degrees 113-146%. Over all sample data points this makes an average of 119%, 129% and 134% relative growth for 10 degree, 20 and 30, respectively (according to Table S1). Thus, there is only a very small difference (less than 5%) between 20 and 30 degrees of growth improvement of CQWZ over SXCG at 30 degrees. Additionally, at 7 out of the 14 data points the relative growth improvement of CQWZ is highest at 20 degrees not at 30 degrees. Thus, there is no relative growth difference regarding temperature (at least at 20 vs 30) between the 2 strains.

Considering the small differences of the average and the variability of the data including measurement variability of 0.43 up to 18.3 % at the different time points. I don’t think there is a significant better growth of CQWZ at 30 degrees compared to 20 degrees and relative to SXCG.

I’m afraid without additional strains showing that the SXCG strain and their growth pattern at different temperatures actually represents the majority of the population and CQWZ is a significant outlier regarding temperature tolerance the presented data don’t warrant the conclusion of the manuscript.

Response: We are very grateful to Reviewer for reviewing the paper so carefully. Citrus target spot was first discovered in the northernmost citrus-producing areas of China (Chenggu County, Shaanxi Province). And was formerly considered a cold-tolerant fungal disease. The disease was only prevalent in late winter and early spring. In recent years, this disease has caused serious damage in the citrus producing areas of Chongqing. Strains from the two regions showed differences in temperature adaptability. Temperatures in both regions do not rise to 30 degrees during an epidemic of the disease. The optimal temperature for the growth of most fungi that cause citrus diseases is between 20 and 25 degrees [1-4]. Under high temperature conditions, the growth rates of all strains are slow down [5]. However, under 30 degrees, the growth rate of CQWZ is still higher than that of SXCG. In the first ten days of strain growth, the average growth under 30 degrees was 113-146%, which was higher than that under 10 degrees (100-116%) and 20 degrees (101-121%) (see Figure 1c in the revised version). We believe these observations are enough to show that CQWZ has a stronger tolerance to high temperature than SXCG.

  1. Everett, K. R. (2003). The effect of low temperatures on Colletotrichum acutatum and Colletotrichum gloeosporioides causing body rots of avocados in New Zealand. Australasian Plant Pathology, 32, 441-448.
  2. Acheampong, M. A., Coombes, C. A., Moore, S. D., & Hill, M. P. (2020). Temperature tolerance and humidity requirements of select entomopathogenic fungal isolates for future use in citrus IPM programmes. Journal of invertebrate pathology, 174, 107436.
  3. Plaza, P., Usall, J., Teixidó, N., & Viñas, I. (2004). Effect of water activity and temperature on competing abilities of common postharvest citrus fungi. International Journal of Food Microbiology, 90(1), 75-82.
  4. Nemec, S., & Zablotowicz, R. M. (1981). Effect of soil temperature on root rot of rough lemon caused by Fusarium solani. Mycopathologia, 76(3), 185-190.
  5. Pietikäinen, J., Pettersson, M., & Baath, E. (2005). Comparison of temperature effects on soil respiration and bacterial and fungal growth rates. FEMS microbiology ecology, 52(1), 49-58.